# Trends of syphilis in Brazil: A growth portrait of the treponemic epidemic

**Marquiony Marques dos Santos** [1,2,3]*, **Ana Karla Bezerra Lopes**[1,4,2,3], **Angelo Giuseppe Roncalli**[1,5], **Kenio Costa de Lima**[1,2,5]

1 Universidade Federal do Rio Grande do Norte, Natal, Rio Grande do Norte, Brazil, 2 Programa de pós-graduação em Ciências da Saúde, Universidade Federal do Rio Grande do Norte, Natal, Rio Grande do Norte, Brazil, 3 Doutorado em Ciências da Saúde, Universidade Federal do Rio Grande do Norte, Natal, Rio Grande do Norte, Brazil, 4 Laboratório de Microbiologia, Maternidade Escola Januário Cicco, Universidade Federal do Rio Grande do Norte, Natal, Rio Grande do Norte, Brazil, 5 Programa de pós-graduação em Saúde Coletiva, Universidade Federal do Rio Grande do Norte, Natal, Rio Grande do Norte, Brazil

* marquiony@gmail.com

**Data Availability Statement:** All five files are available from the Figshare database (accession: https://doi.org/10.6084/m9.figshare.11799780)

**Funding:** This research was funded by the Ministry of Health, and the Laboratório de Inovação

## Abstract

Syphilis is a chronic infectious disease with its prevalence being described since the 15th century. Although its etiological agent and also the treatment measures are widely known, syphilis is still a great public health problem worldwide, mainly in countries with limited resources associated to low investments in health primary care. The aim of the present study was to analyze the trend and regional distribution of syphilis in Brazil between 2007 and 2017. This is an ecological study using secondary data from the Brazilian notification system. The Ministry of Health selected 100 municipalities which presented the worse outcomes related to syphilis from the 5,570 Brazilian municipalities as a target for a comprehensive project in order to tackle the prevalence of syphilis, called the "No Syphilis Project". These priority municipalities represent 57.7% of syphilis cases and about one third of the Brazilian population. They were compared with other 189 non-priority municipalities with more than 100 thousand inhabitants among the Brazilian regions (North, Northeast, Southeast, South and Center-West). Polynomial regression methods and Joinpoint analyses were used to analyze the trend, from which the Annual Average Percent Change (AACP) for each time period was calculated. There was a significant growth trend in all regions for the main three forms of syphilis (in pregnancy, congenital and acquired), especially in the South. The ratio between syphilis in pregnancy and congenital syphilis increased in both priority (AACP: 8.54%; $p<0.001$) and non-priority municipalities (AACP: 2.61%; $p = 0.005$), as well as in the regions, except the Center-West. High growth trends in syphilis prevalence were found in all municipalities, as well as all five regions between 2007 and 2017, showing that the challenge to reduce or even eliminate syphilis in Brazil is still difficult.

## Introduction

Syphilis is a chronic infectious disease with its prevalence being described since the 15th century. Although its etiological agent and also the treatment measures are widely known [1], syphilis is still a great public health problem worldwide, mainly in countries with limited

Tecnológica em Saúde (LAIS/HUOL/UFRN) process number 732017, Project: "Syphilis No." The funders had no role in study design, data collection and analysis, decision to publish, or preparation of the manuscript.

**Competing interests:** The authors have declared that no competing interests exist.

resources associated to low investments in health primary care. Therefore, although being a relatively simple disease, it keeps challenging the most complex public health systems [2].

Syphilis is among the most common sexually transmitted diseases (STD), mostly being transmitted by unprotected sexual practices and in pregnancy from the mother to the foetus. Syphilis can progress into more severe stages in the absence of treatment, from the primary syphilis to secondary or tertiary, and it also presents a latency period [1,3].

Thus, several stages in primary, secondary and latent syphilis and also the different strategies of both diagnostic and treatment toward different population groups lead to syphilis being a disease which is very difficult to eradicate. The efficacy of these strategies depends on good organization of the healthcare systems, which is not usual in developing countries [1,2].

Several countries are dealing with an increase in syphilis cases, which are affecting different population groups. Regarding syphilis in pregnancy, there is an estimate of 1.36 million women with active syphilis in all countries with available data, along with 91,764 neonatal deaths in 2008 [2]. About three million new cases of syphilis were registered in 2008 in the Latin America and Caribbean regions, with 937 thousand among these being in Brazil [4].

According to the Brazilian Ministry of Health, an increase in registered syphilis cases was observed in its 2018 annual report over the last ten years. In 2017, 119,800 cases of acquired syphilis were notified, meaning a rate of 58.1 cases per 100 thousand inhabitants, which is considered very high [5].

Regarding syphilis in pregnancy, 49,013 cases were notified in this same year, which means a rate of 17.2 per thousand live births. For congenital syphilis, 24,666 cases were registered, meaning a rate of 8.6 cases per thousand live births [5].

Nowadays, Brazil is divided into five regions based on geographical features (North, Northeast, Southeast, South and Center-West) and 5,570 municipalities. The distribution of syphilis among these regions and municipalities is highly heterogenous and usually municipalities with a larger population have higher probabilities to present an epidemic profile for syphilis compared to other municipalities.

Thus, it is necessary to identify the trends in syphilis in different locations and considering all types of its manifestation, i.e. syphilis in pregnancy, congenital and acquired, as they have specific transmission forms and also different prevention and treatment strategies. Regarding acquired syphilis, this type is usually neglected and there are few studies available [3].

The occurrence of syphilis in its different forms can be a predictor of important failures in the health services and its increase has raised national concern in recent years. Therefore, it is crucial to analyse the characteristics of the trend in syphilis rates, whose results could guide new public health policies and improve the current ones, enabling monitoring health indicators toward strategic planning, mainly concerning the assessment of interventions focused on reducing syphilis indicators. Thus, the aim of this study is to analyse the trend of acquired syphilis between 2011 and 2017, syphilis in pregnancy and congenital syphilis between 2007 and 2017, and also the ratio between syphilis in pregnancy and congenital syphilis considering regions and municipalities of Brazil.

## Methods

This study is part of a major effort to reduce the syphilis epidemic throughout a nationwide project named "Applied research for intelligent integration oriented to strengthening the healthcare networks for a rapid response to syphilis", also called "Syphilis No", as a short name. The main objective of this project is to reduce acquired syphilis and to also eliminate congenital syphilis in Brazil [6]. Several measures have been implemented to achieve this goal,

with the development of studies implementing different designs among them in order to subsidize strategies to tackle the syphilis epidemic.

Thus, in an attempt to contribute to improving knowledge about syphilis in Brazil, an ecological trend study was developed to understand the behavior of the main types of syphilis, using the following periods: (a) the period from 2011 to 2017 was used for the detection rate of acquired syphilis, the former because it represents the beginning of the official notification in Brazil, and the latter because it is the last year with available data; (b) the period used regarding rates of congenital syphilis and in pregnancy was 2007 to 2017, for the same reasons as mentioned above.

## Data sources

Data on syphilis were gathered from the official health information systems of the Brazilian Ministry of Health. Among these systems, the "Information System of Mandatory Notification Diseases" (SINAN, from the Portuguese acronym) is the main source of data on syphilis, as all cases of this disease are notified and registered in this system. The Information System of Live Births (SINASC from the Portuguese acronym) was also used to calculate the detection rates of syphilis, and data from the National Institute of Geography and Statistics (IBGE, from the Portuguese acronym) was included for socioeconomic variables.

## Calculation of the rate detection

According to the Brazilian Ministry of Health, detection rates of the different types of syphilis must be calculated from the instructions shown in Box 1.

## Unit of analysis

The municipalities included in this study were chosen from criteria established by the Brazilian Ministry of Health as a strategy to prioritize those that would be included in a more

Box 1. Calculation of detection rate of acquired syphilis, detection rate of syphilis in pregnancy, incidence rate of congenital syphilis in children aged under one year and ratio between syphilis in pregnancy and congenital syphilis.

| Indicator | Calculation | | Sources |
|---|---|---|---|
| Detection rate of acquired syphilis | Number of cases of acquired syphilis in individuals aged 13 years and over in such a year of diagnostic | x100,000 | SINAN |
| | Population of individuals aged 13 years and over in the same year | | IBGE |
| Detection rate of syphilis in pregnancy | Number of cases of syphilis detected in pregnant women in such a year of diagnostic | x1000 | SINAN |
| | Number of live births from mothers in the same local of residence and year of diagnostic | | SINASC |
| Incidence rate of congenital syphilis in children aged under one year | Number of new cases of congenital syphilis in children aged under one year in such a year of diagnostic | x1000 | SINAN |
| | Number of live births from mothers in the same local of residence and year of diagnostic | | SINASC |
| Ratio between syphilis in pregnancy and congenital syphilis | Detection rate of syphilis in pregnancy | - | SINAN / SINASC |
| | Incidence rate of congenital syphilis in children aged under one year | | SINAN / SINASC |

comprehensive amount of health measures, focusing on health networks. In order to do this, a ranking was adopted taking into account the size of population and a compound index, formed by some key indicators of syphilis, such as incidence of congenital syphilis in the last five years and its respective percentage increase, and the perinatal mortality rate and its respective percentage increase.

In addition to the compound index, the prioritization follows populational criteria, meaning that the capital cities and municipalities in metropolitan areas were included as priority 1 and 2, respectively. The remaining municipalities were defined as priority 3. The 100 priority cities represent 31% of the whole Brazilian population (64,271,031 inhabitants) and concentrate 57.7% of the amount of cases of syphilis in Brazil in 2015. The remaining municipalities with more than 100,000 inhabitants (186) were considered as non-priority in this study.

The syphilis trend was analyzed in order to identify whether a possible increase, steadiness or decline over the studied period would be significant or there was a random effect. Data were analyzed through two different regression methods, namely polynomial and log-linear. The polynomial regression model was chosen because it is easy to interpret, as well as for its high statistical power [7]. The log-linear model was necessary to know the weighted average of the inflection points that would be present overtime, and also to verify the statistical significance of these changes. The polynomial regression model was started from a simpler model with a grade one linear function ($Y = \beta_0 + \beta_1 X_1$). The simplest model remained in the results, which could explain its adequacy through the force of the adjustment of the regression line and also from both the coefficient of determination ($R^2$) and the statistical significance (p-value below 0.05).

The log-linear regression model was carried out using the Joinpoint software program (version 4.7.0.0), in which any point of change in the line was assessed in order to verify the statistical significance. The Average Annual Percent Change (AACP) was calculated from the weighted average of the slope coefficients in the subjacent regression line to summarize all variation which happened in studied period [8], with weights equal to the measured interval.

Since this studied used only secondary data available in public databases, it was not necessary to submit the research proposal to an ethical committee.

## Results

The rate of acquired syphilis in Brazil presented a substantial increase, from 12.3 cases per 100,000 inhabitants in 2011 to 81.4 in 2017, which means a gross growth rate of 561%. Fig 1 depicts the syphilis trend in pregnancy, where we can see an increase from 2.2 per 1,000 live births in 2007 to 16.9 in 2017, meaning a gross growth rate of 660%. Finally, congenital syphilis presented a rate of 2.00 per 1,000 live births in 2007, jumping to 8.8 in 2017 (gross growth rate of 338%).

In analyzing the trend in the results related to acquired syphilis from 2011 to 2017 (Table 1), a high and significant growth trend in both priority (AAPC: 30.9; p<0.001) and non-priority municipalities (AAPC: 36.2; p<0.001) can be seen. Regarding regions of the country, the South region presented a significant increase of 60.4% (p<0.001).

Table 2 illustrates a strong increasing trend in syphilis in pregnancy in priority (AAPC: 25.5%; p<0.001) and non-priority (23,6%; p<0.001) municipalities, as well as in the Brazilian regions, especially in the southern region (30.8%; p<0.001).

The trend in congenital syphilis can be observed in Table 3, which is quite similar to previous ones. Priority municipalities presented an AAPC of 15.7% (p<0.001), while the non-priority municipalities presented 20.4% (p<0.001). Among regions, the South presented the highest value (AAPC: 26.0; p<0.001).

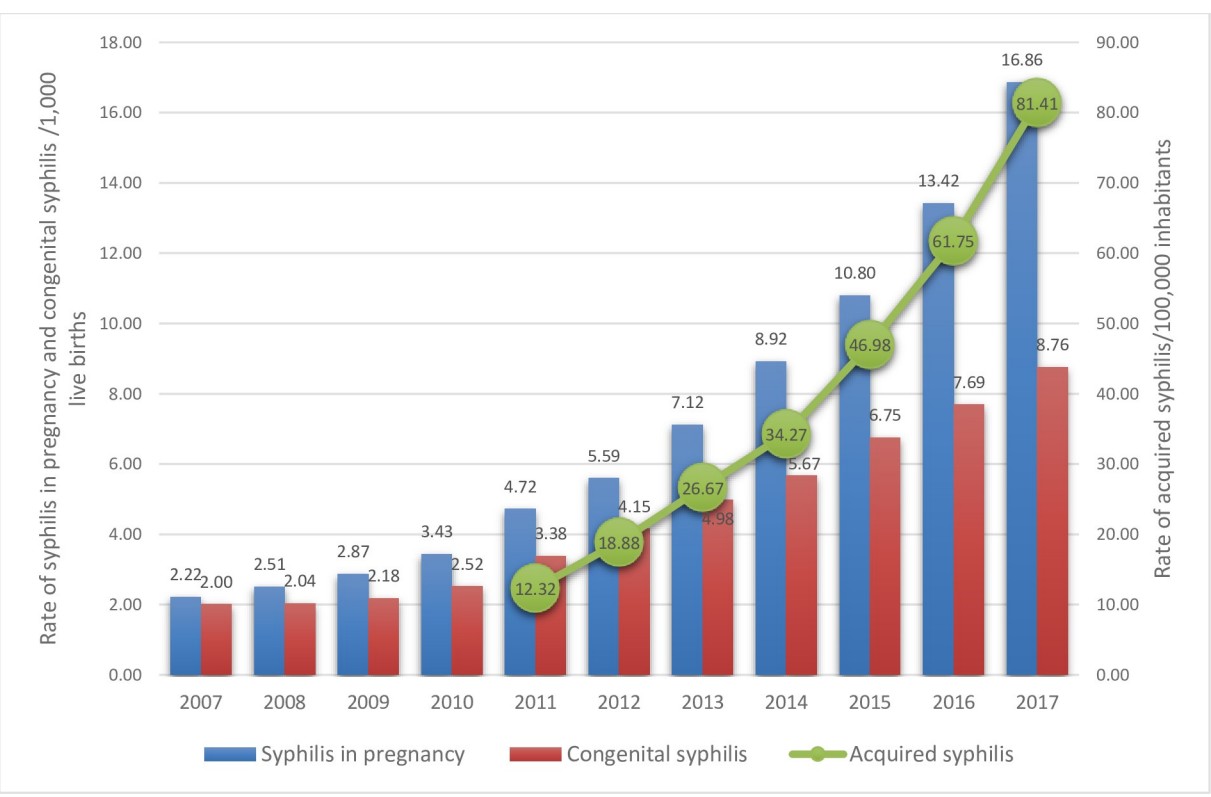

**Fig 1. Distribution of syphilis in pregnancy rate, congenital syphilis rate and acquired syphilis rate in Brazil.** Database: National System of Aggravations and Notification—SINAN (2018).

Finally, regarding the ratio between syphilis in pregnancy and congenital syphilis (Table 4), a growing trend was observed in all groups (type of municipality and regions). The constant in the Northeast region was 0.92, which means that the congenital syphilis rate is higher than syphilis in pregnancy, showing a significant increase with an AAPC of 1.94% (p<0.001). The Center-West region presented a reduction of 4.72 (p = 0.001), meaning an approximation between the number of syphilis cases in pregnancy and congenital syphilis.

**Table 1. Trend analysis of acquired syphilis rate in priority and non-priority municipalities and according to regions of Brazil, from 2011 to 2017.** Database: National System of Aggravations and Notification—SINAN (2018).

| | $Y = \beta_0 + \beta_1 X_1$ | | | AAPC (%) | *p*-value |
|---|---|---|---|---|---|
| | $\beta_0$ | $\beta_1$ | $R^2$ | | |
| Type of municipality | | | | | |
| *Priority* | 60.91 | 14.92 | 0.98 | 30.91 | <0.001 |
| *Non-Priority* | 42.99 | 11.94 | 0.96 | 36.17 | <0.001 |
| Region | | | | | |
| *North* | 21.02 | 7.12 | 0.86 | 44.93 | <0.001 |
| *Northeast* | 16.98 | 5.57 | 0.87 | 40.81 | <0.001 |
| *Southeast* | 53.32 | 12.07 | 0.98 | 27.95 | <0.001 |
| *South* | 58.58 | 21.07 | 0.96 | 60.38 | <0.001 |
| *Center-West* | 29.64 | 10.06 | 0.89 | 44.53 | <0.001 |

AAPC: *Average Annual Percent Change.*

**Table 2. Trend analysis of syphilis in pregnancy rate in priority and non-priority municipalities and according to regions of Brazil, from 2007 to 2017.** Database: National System of Aggravations and Notification—SINAN (2018).

| | $Y = \beta_0 + \beta_1 X_1$ | | | AAPC (%) | *p*-value |
|---|---|---|---|---|---|
| | $\beta_0$ | $\beta_1$ | $R^2$ | | |
| Type of municipality | | | | | |
| *Priority* | 10.04 | 2.04 | 0.93 | 25.65 | <0.001 |
| *Non-Priority* | 6.35 | 1.27 | 0.90 | 23.57 | <0.001 |
| Region | | | | | |
| *North* | 6.95 | 1.10 | 0.84 | 16.33 | <0.001 |
| *Northeast* | 5.01 | 0.84 | 0.92 | 19.06 | <0.001 |
| *Southeast* | 8.31 | 1.79 | 0.93 | 28.42 | <0.001 |
| *South* | 7.82 | 1.88 | 0.88 | 30.78 | <0.001 |
| *Center-West* | 7.89 | 1.04 | 0.85 | 13.44 | <0.001 |

AAPC: *Average Annual Percent Change.*

## Discussion

In analyzing the syphilis trends in its different forms in Brazil, a significant high growing trend was found, especially in acquired syphilis, in all regions and both types of municipalities. The ratio between syphilis in pregnancy and congenital syphilis had a similar pattern with an exception for the Center-West region, which presented a significant negative trend.

The strong growing trend in acquired syphilis in Brazil must be analyzed with caution, because acquired syphilis only had its notification mandatory in the whole country after 2010 [9]. Thus, the growing trend could be caused by the increase in the number of notifications. However, several studies have demonstrated an increase in acquired syphilis worldwide, including Latin America [4,10]. In this context, the starting of compulsory notification cannot be pointed out as the only reason, even because the growing trend remains steady in the period from 2011 to 2017.

One important factor that must be implicated in the growing trend in acquired syphilis is the reduction in the use of condoms. A study carried out by Stover et al. [11] assessed the estimates on the use of male condoms in 81 countries, as well as its cost-effectiveness in reducing sexually transmitted diseases. The authors concluded that governments must expand the

**Table 3. Trend analysis of congenital syphilis rate in priority and non-priority municipalities and according to regions of Brazil, from 2007 to 2017.** Database: National System of Aggravations and Notification—SINAN (2018).

| | $Y = \beta_0 + \beta_1 X_1$ | | | AAPC (%) | *p*-value |
|---|---|---|---|---|---|
| | $\beta_0$ | $\beta_1$ | $R^2$ | | |
| Type of municipality | | | | | |
| *Priority* | 7.92 | 1.10 | 0.97 | 15.75 | <0.001 |
| *Non-Priority* | 3.21 | 0.58 | 0.91 | 20.43 | <0.001 |
| Region | | | | | |
| *North* | 3.59 | 0.44 | 0.81 | 12.04 | <0.001 |
| *Northeast* | 4.96 | 0.71 | 0.97 | 16.78 | <0.001 |
| *Southeast* | 4.94 | 0.77 | 0.96 | 18.13 | <0.001 |
| *South* | 4.22 | 0.88 | 0.93 | 26.28 | <0.001 |
| *Center-West* | 3.04 | 0.51 | 0.93 | 19.06 | <0.001 |

AAPC: *Average Annual Percent Change.*

**Table 4. Trend analysis of the ratio between syphilis in pregnancy and congenital syphilis in priority and non-priority municipalities and according to regions of Brazil, from 2007 to 2017.** Database: National System of Aggravations and Notification—SINAN (2018).

| | $Y = \beta_0 + \beta_1 X_1 + \beta_2 X_2$ | | | | AAPC (%) | *p*-value |
|---|---|---|---|---|---|---|
| | $\beta_0$ | $\beta_1$ | $\beta_2$ | $R^2$ | | |
| Type of municipality | | | | | | |
| *Priority* | 1.14 | 0.09 | NA | 0.97 | 8.54 | <0.001 |
| *Non-Priority* | 1.90 | 0.05 | NA | 0.61 | 2.61 | 0.005 |
| Region | | | | | | |
| *North* | 1.85 | 0.07 | NA | 0.70 | 3.82 | <0.001 |
| *Northeast* | 0.92 | 0.02 | 0.01 | 0.79 | 1.94 | <0.001 |
| *Southeast* | 1.50 | 0.12 | NA | 0.96 | 8.69 | <0.001 |
| *South* | 1.57 | 0.06 | 0.01 | 0.88 | 3.56 | <0.001 |
| *Center-West* | 2.81 | -0.14 | NA | 0.73 | -4.72 | 0.001 |

AAPC: *Average Annual Percent Change.*

NA: *Not Available.*

programs focusing on prevention and use of male condoms, because the impact of these measures are very significant, especially in populations with at-risk behavior, such as men who have sex with men (MSM) and sex workers.

Although our study did not assess the syphilis trends in populations with at-risk behaviors, these groups have been highlighted regarding the acquired syphilis infection, showing a strong increase. An important increase in acquired syphilis in MSM was observed in the United States and Eastern Europe starting from 2000, and this increase can be explained by the at-risk sexual behaviors, such as anal sex without a condom, multiple sexual partners, use of drugs before sex, and high risk anonymous sexual contacts, mainly after using smartphone dating apps [4,10]. Therefore, the high growing rates of acquired syphilis presented in our study can be also caused by the change in the behavior of these key populations, mainly in the studied municipalities, which are capital cities and also have a large population size.

There were significant growth trends for syphilis in pregnant women and in congenital syphilis in both priority and non-priority municipalities and also in the Brazilian regions in our study. Several studies corroborate these findings, pointing to syphilis in the gestational period and at birth as a serious public health problem due to its high prevalence [12–15].

A study developed in six federative units (states) in Brazil[14] found a significant increase in syphilis in pregnancy in five states from 2007 to 2012. The state of Amazonas, in the North region, presented a growth rate of 21.2% and the highest rate was found in the state of Rio de Janeiro, in the Southeast region (63.9%) [16].

Nowadays Brazil is facing an epidemic of syphilis in comparison with other countries in Latin America, following an opposite trend in the goal to eradicate congenital syphilis, as it has presented a strong growing trend in the incidence rates in the last 10 years [17]. The increase of syphilis in pregnancy rates evidences an important failure in prenatal assistance in Brazilian primary healthcare services. According to Nonato, Melo and Guimarães [12], there is a significant association between congenital syphilis and some conditions of the mothers, such as an age below 20, low education, tobacco use, late prenatal start, less than six prenatal consultations and not having been tested for syphilis in the first three months of pregnancy. Thus, the quality of care in the prenatal, birth and puerperal periods are important elements for tackling syphilis in pregnancy and congenital syphilis, requiring a proper structure of health services, especially in primary healthcare.

The treponemic rapid test for detecting syphilis in primary healthcare is an important component of prenatal care, helping professionals to accomplish adequate diagnosis and treatment [17–19]. Although the Brazilian Ministry of Health make these tests available in prenatal care offered in public primary healthcare facilities [20], it seems that their utilization is still not broadly incorporated into the healthcare practice on a daily basis.

Lazarini and Barbosa [21] developed a study based on an educational intervention on diagnosis, treatment and notification with 102 primary healthcare professionals. They found that 92.2% of the professionals did not know about the correct procedure after a positive result in a reagent non-treponemal test before the intervention. Therefore, the investigation of syphilis in pregnancy may be underreported and the trends presented in our study could be even larger, especially because the population generally presented a high trend in acquired syphilis.

Regarding the ratio between syphilis in pregnancy and congenital syphilis, significant growing trends in both priority and non-priority municipalities were found, and in all regions except the Center-West region. This result means that syphilis detection in pregnant women is higher than the detection congenital syphilis, and this trend is ascending. Although the early detection of the treponemic contamination in infants is a difficult diagnosis, mainly in an asymptomatic phase, the improvement in diagnosing it in prenatal care would facilitate the struggle against syphilis [13].

However, there was a significant reduction in the trend of the ratio between syphilis in pregnancy and congenital syphilis in the Center-West region. This scenario can indicate that number of pregnant women diagnosed with syphilis becomes closer to the number of congenital syphilis, which could mean a late diagnosis of syphilis in pregnancy as well as a lack of treatment [12].

The ratio between syphilis in pregnancy and congenital syphilis in the Northeast region showed a significant trend and with a higher congenital syphilis rate than for syphilis in pregnancy. It is recommended that the detection rate of syphilis in pregnancy be higher than congenital syphilis, which means that the health services are being able to diagnose more pregnant women and could more effectively intervene in congenital syphilis. Thus, this measure can indirectly evaluate the local quality of health care [22].

The significant growing trend in the ratio between syphilis in pregnancy and congenital syphilis in the Northeast region may be explained by some hypotheses. One of them would be failures in the use of diagnostic and treatment algorithms related to pregnant women and new-born children by the health professionals, besides a more fragile structure in health care services in this specific region.

Another hypothesis is related to the late treatment after a diagnosis obtained in hospital, whose criteria for treatment and notification are different from those used in primary healthcare. A study conducted in Northeast Brazil [23] demonstrates a high prevalence in the inadequate treatment of syphilis in pregnancy, which could explain the growing trends found in our study.

The limitations of our study are related to the aforementioned underreporting of syphilis cases in pregnancy and in congenital syphilis. Another important point is the compulsory notification of acquired syphilis that only started from 2010, which could oversize the growth in the first years from 2010. In an attempt to overcome this bias, our study only included data from 2011. Another important aspect to be considered is the quality of data, as secondary data from health services are influenced by the health surveillance services; however, it is noteworthy that these strategies are showing improvements in recent years. Another important remark to be considered is the difficulty in adequately diagnosing congenital syphilis in asymptomatic infants born to seropositive mothers, although diagnostic algorithms have been improved over the last years.

We conclude that there are significant growing trends in syphilis in its different forms in Brazil, with a great risk to public health. The trend in acquired syphilis may indicate a more expressive increase in syphilis in pregnancy in the following years. Therefore, enhancing the investments in prevention and health promotion must be a priority, especially those measures which focus on the Healthcare Networks, in both ways: improving the quality of pregnancy care and the healthcare target to populations with at-risk behaviors.

Treating a disease that affects different populations, such as syphilis, demands more effective and integrated measures of health services in all levels of complexity. Therefore, in order to improve the access and the quality in prenatal care, mainly with measures focused on training primary healthcare professionals to increase the offer of treponemic tests in the first prenatal consultation and also developing intervention programs of targeted to vulnerable groups are necessary strategies in the attempt to reduce the trends in syphilis demonstrated in this study. The commitment of eliminating syphilis is still a great challenge in Brazil, whose adopted measures must be evaluated by observing their capacity to modify the current scenario.

## Supporting information

**S1 Fig. Distribution of syphilis in pregnancy rate, congenital syphilis rate and acquired syphilis rate in Brazil.**
(XLSX)

**S1 Table. Polynomial regression and log-linear regression between acquired syphilis rate in 2011 to 2017 in priority and non-priority municipalities and according to regions of Brazil.**
(XLSX)

**S2 Table. Polynomial regression and log-linear regression between syphilis in pregnancy rate in 2007 to 2017 in priority and non-priority municipalities and according to regions of Brazil.**
(XLSX)

**S3 Table. Polynomial regression and log-linear regression between congenital syphilis rate in 2007 to 2017 in priority and non-priority municipalities and according to regions of Brazil.**
(XLSX)

**S4 Table. Polynomial regression and log-linear regression of the ratio between syphilis in pregnancy and congenital syphilis in 2007 to 2017 in priority and non-priority municipalities and according to regions of Brazil.**
(XLSX)

**S1 Checklist. STROBE statement—checklist of items that should be included in reports of *cross-sectional studies*.**
(DOCX)

## Author Contributions

**Conceptualization:** Marquiony Marques dos Santos, Ana Karla Bezerra Lopes, Angelo Giuseppe Roncalli, Kenio Costa de Lima.

**Formal analysis:** Marquiony Marques dos Santos.

**Funding acquisition:** Angelo Giuseppe Roncalli, Kenio Costa de Lima.

**Investigation:** Marquiony Marques dos Santos, Ana Karla Bezerra Lopes, Angelo Giuseppe Roncalli, Kenio Costa de Lima.

**Methodology:** Marquiony Marques dos Santos, Angelo Giuseppe Roncalli, Kenio Costa de Lima.

**Resources:** Angelo Giuseppe Roncalli, Kenio Costa de Lima.

**Supervision:** Marquiony Marques dos Santos, Kenio Costa de Lima.

**Writing – original draft:** Marquiony Marques dos Santos.

**Writing – review & editing:** Ana Karla Bezerra Lopes, Angelo Giuseppe Roncalli, Kenio Costa de Lima.

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
