## [Decision Letter · Decision Letter 0]

4 Dec 2019

PONE-D-19-31379

Trends of syphilis in Brazil: a growth portrait of the treponemic epidemic

PLOS ONE

Dear Dr. Marques dos Santos,

Thank you for submitting your manuscript to PLOS ONE. After careful consideration, we feel that it has merit but does not fully meet PLOS ONE’s publication criteria as it currently stands. Therefore, we invite you to submit a revised version of the manuscript that addresses the points raised during the review process.

We would appreciate receiving your revised manuscript by Jan 18 2020 11:59PM. To enhance the reproducibility of your results, we recommend that if applicable you deposit your laboratory protocols in protocols.io, where a protocol can be assigned its own identifier (DOI) such that it can be cited independently in the future. For instructions see: http://journals.plos.org/plosone/s/submission-guidelines#loc-laboratory-protocols

We look forward to receiving your revised manuscript.

Kind regards,

Zhefeng Meng, M.D., Ph.D.

Academic Editor

PLOS ONE

Journal Requirements:

1. We suggest you thoroughly copyedit your manuscript for language usage, spelling, and grammar. If you do not know anyone who can help you do this, you may wish to consider employing a professional scientific editing service.  

2. Please include a copy of Table 5 which you refer to in your text on page 12.

3. We note you have included a table to which you do not refer in the text of your manuscript. Please ensure that you refer to Table 4 in your text; if accepted, production will need this reference to link the reader to the Table.

Reviewers' comments:

Reviewer's Responses to Questions

**Comments to the Author**

1. Is the manuscript technically sound, and do the data support the conclusions?

Reviewer #1: Partly

2. Has the statistical analysis been performed appropriately and rigorously? 

Reviewer #1: Yes

3. Have the authors made all data underlying the findings in their manuscript fully available?

Reviewer #1: Yes

4. Is the manuscript presented in an intelligible fashion and written in standard English?

Reviewer #1: No

5. Review Comments to the Author

Reviewer #1: The manuscript by Marques dos Santos et al. “Trends of syphilis in Brazil: a growth ” addresses an important situation of the increase in acquired/congenital syphilis infections in Brazil.

Comments:

1. Abstract, Page 2, Lines 30-31: The sentence “In order to analyze the trend and regional distribution of syphilis in Brazil between 2007-2017.” Is an incomplete sentence.

2. Page 3, Line 53: syphilis diagnosis is complex, this should be reworded. Similarly, page 3, Line 70: syphilis has been termed “the great mimicker” which is because the disease is complex in that it mimics many other diseases. It is suggested this be reworded as well, perhaps to emphasize that the disease is easy to treat if appropriately diagnosed.

3. Page 3, Line 71: the wording of this sentence should be revised, suggest “Syphilis is among the…”. The meaning of “contamination by syphilis bacteria” is unclear. Similarly, Page 14, Line 254, the word “contamination” should be removed.

4. Page 4, Lines 81-83: this sentence should be cited by reference 2.

5. Figure 1: the right Y axis should be relabeled for clarity to “rate of acquired syphilis/100,000 inhabitants”.

6. Page 12, Line 221: Table 4 is incorrectly labeled as Table 5.

7. Page 14, Line 268: what do the authors mean by “rightest growth”? This needs to be reworded for clarity.

8. Page 15, Line 289: what is a “reagent non-treponemal test”?

9. Page 16, Lines 295-297: This statement is speculative. The fact that the ratio determination has indicated that syphilis detection in pregnant women is higher than congenital syphilis does not necessarily translate to a higher number of pregnant women being detected in prenatal care. It could also indicate that more infants were born with asymptomatic congenital syphilis.

10. Related to the point above, how has the methodology accounted for asymptomatic congenital syphilis versus no evidence of congenital syphilis (either symptomatic or asymptomatic)? There is significant difficulty in adequately diagnosing congenital syphilis in asymptomatic infants born to seropositive mothers. The authors should note this as a limitation of their study.

11. Page 16, Lines 298-301: this trend could also indicate early diagnosis but lack of treatment. This possibility should be stated.

12. Page 16, Lines 314-315: A broader discussion of the health care services available in the study regions, in relation to the ratio of syphilis in pregnancy/congenital syphilis, would be informative and could uncover additional reasons for the observed trends.

6. PLOS authors have the option to publish the peer review history of their article (what does this mean?). If published, this will include your full peer review and any attached files.

Reviewer #1: No

---

## [Author Response · Author response to Decision Letter 0]

14 Feb 2020

RESPONSE TO REVIEWERS

Journal Requirements:

1. Please ensure that your manuscript meets PLOS ONE's style requirements: 

The authors have double checked the paper and it meets the requirements.

2. Professional scientific editing service:

It was sent a document with the confirmation of the professional editing service by ECB - English Consulting Brazil, sponsored by Christopher J. Quinn.

3. Please include a copy of Table 5 which you refer to in your text on page 12.

There is no Table 5, it is actually Table 4. It is now correct in the paper.

4. We note you have included a table to which you do not refer in the text of your manuscript. Please ensure that you refer to Table 4 in your text; if accepted, production will need this reference to link the reader to the Table.

The correction has been made.

Comments to the Author

Our conclusions were based in the results presented in page 17, lines 324 to 337. We have included a synthesis of these results confronted with the findings of updated scientific literature. In our opinion, the conclusions are appropriately based on the data presented.

2. Has the statistical analysis been performed appropriately and rigorously?

No further actions needed.

3. Have the authors made all data underlying the findings in their manuscript fully available?

No further actions needed.

4. Is the manuscript presented in an intelligible fashion and written in standard English?

No further actions needed. It was sent a document with the confirmation of the professional editing service by ECB - English Consulting Brazil, sponsored by Christopher J. Quinn.

5. Review Comments to the Author.

All suggested modifications have been made. The paper was rigorously revised and the file with modifications was sent.

6. 6. PLOS authors have the option to publish the peer review history of their article (what does this mean?). If published, this will include your full peer review and any attached files.

No further actions needed.

---

## [Editor Report · Decision Letter 1]

16 Mar 2020

Trends of syphilis in Brazil: a growth portrait of the treponemic epidemic

PONE-D-19-31379R1

Dear Dr. Marques dos Santos,

We are pleased to inform you that your manuscript has been judged scientifically suitable for publication and will be formally accepted for publication once it complies with all outstanding technical requirements.

With kind regards,

Zhefeng Meng, M.D., Ph.D.

Academic Editor

PLOS ONE
---

## [Editor Report · Acceptance letter]

23 Mar 2020

PONE-D-19-31379R1 

Trends of syphilis in Brazil: a growth portrait of the treponemic epidemic 

Dear Dr. Marques dos Santos:

I am pleased to inform you that your manuscript has been deemed suitable for publication in PLOS ONE. Congratulations! Your manuscript is now with our production department. 

With kind regards,

on behalf of

Dr. Zhefeng Meng 

Academic Editor

PLOS ONE